# A Solar Irradiance Forecasting Framework Based on the CEE-WGAN-LSTM Model

**DOI:** 10.3390/s23052799

**Published:** 2023-03-03

**Authors:** Qianqian Li, Dongping Zhang, Ke Yan

**Affiliations:** 1Key Laboratory of Electromagnetic Wave Information Technology and Metrology of Zhejiang Province, College of Information Engineering, China Jiliang University, Hangzhou 310018, China; 2Department of the Built Environment, College of Design and Engineering, National University of Singapore, 4 Architecture Drive, Singapore 117566, Singapore

**Keywords:** time-series forecasting, deep learning, signal decomposition, solar irradiance

## Abstract

With the rapid development of solar energy plants in recent years, the accurate prediction of solar power generation has become an important and challenging problem in modern intelligent grid systems. To improve the forecasting accuracy of solar energy generation, an effective and robust decomposition-integration method for two-channel solar irradiance forecasting is proposed in this study, which uses complete ensemble empirical mode decomposition with adaptive noise (CEEMDAN), a Wasserstein generative adversarial network (WGAN), and a long short-term memory network (LSTM). The proposed method consists of three essential stages. First, the solar output signal is divided into several relatively simple subsequences using the CEEMDAN method, which has noticeable frequency differences. Second, high and low-frequency subsequences are predicted using the WGAN and LSTM models, respectively. Last, the predicted values of each component are integrated to obtain the final prediction results. The developed model uses data decomposition technology, together with advanced machine learning (ML) and deep learning (DL) models to identify the appropriate dependencies and network topology. The experiments show that compared with many traditional prediction methods and decomposition-integration models, the developed model can produce accurate solar output prediction results under different evaluation criteria. Compared to the suboptimal model, the MAEs, MAPEs, and RMSEs of the four seasons decreased by 3.51%, 6.11%, and 2.25%, respectively.

## 1. Introduction

Renewable energy and green infrastructure are widely considered to be the most critical factors in promoting the sustainable development of intelligent cities, both in the current and in the coming decades [1,2]. Solar energy is experiencing unprecedented development as a sustainable renewable energy, and climate change and environmental pollution are of serious concern [3,4,5]. Meanwhile, electricity conversion technology has significantly reduced investment costs, thus greatly improving the market competitiveness of solar energy. According to the literature, in the past ten years, the cost of solar photovoltaic power has decreased by 85%. The average power of PV modules has increased from 250–300 W in 2010 to 400–550 W in 2020 and is expected to reach 800–1200 W by 2030 [6]. The continuous improvement of PV module power depends on the continuous development of PV technology. Unlike other manageable energy sources, solar energy output has the characteristics of high nonlinearity, strong randomness, and large intermittency because solar radiation intensity is affected by various complex factors. Iqbal et al. [7] found that the environment’s dew point temperature and humidity are inversely proportional to the solar radiation intensity. Mustafa et al. [8] found that many environmental factors (such as dust accumulation, water droplets, bird droppings, and partial shade conditions) have a significant impact on the performance of photovoltaic systems. Therefore, the accurate and reliable prediction of solar irradiance is crucial for promoting large-scale integration of solar energy and improving the operating efficiency of power systems with multiple energy sources [9].

According to the existing literature, scholars have proposed many data-driven prediction methods to predict solar irradiance, which can be roughly divided into two categories: single and mixed models [10]. The single-model prediction methods include traditional statistical, classical machine learning, and deep learning methods [11]. The research identifies an issue with lag in the prediction task of a single model, which cannot sufficiently reflect the randomness of photovoltaic fluctuations. Mixed-model prediction methods usually combine multiple models to solve the limitations of independent models. Using the characteristics of multiple models to improve performance or combining these methods with feature engineering can solve the problem of poor prediction accuracy of a single model [12,13]. One of the leading hybrid-model prediction methods is the decomposition-integration model.

As an effective and robust method, the decomposition-integrated deep learning framework has attracted attention in various fields and has been applied in diverse areas, including wind speeds [14], photovoltaics [15], coal prices [16], agribusiness [17], and electricity prices [18]. On one hand, mode-decomposition technology decomposes irradiance data into several frequency subsequences, which reduces the impact of noise on prediction accuracy and makes it easier for prediction models to capture changing modes [19]. Empirical mode decomposition (EMD), variational mode decomposition (VMD), and wavelet decomposition (WTD) are among the methods most commonly used by scholars for the decomposition preprocessing of photovoltaic data. On the other hand, the aim of the prediction model is to apply an efficient and simple model to predict the decomposition subsequences. Scholars typically adopt common DL models, such as artificial neural networks (ANNs), convolutional neural networks (CNNs), LSTM, gated recursive units (GRUs), and generative adversarial networks (GANs) [20,21]. GANs have become an important branch of the field of DL with their unique confrontation ideas and unsupervised learning methods. Although GANs have been successfully applied to computer vision and achieved good results, they are rarely used in time-series predictions. Because of the excellent ability of GANs to capture the underlying relationship between data, this paper is committed to applying GANs to photovoltaic time-series predictions [22].

Although some studies have confirmed that the hybrid model based on mode-decomposition technology can improve prediction accuracy, some areas still need to be studied and improved. First, considering the combination with mode-decomposition technology, many researchers often use traditional DL prediction models and rarely consider using newer DL prediction models. The second is that most studies only use one model to predict the decomposition subsequences after data decomposition and lack consideration of the spectral differences between the decomposition sequences, that is, the diversity and adaptability of high- and low-frequency data matching the prediction model.

To solve the above problems, this paper combines the methods of previous studies to propose a new decomposition-integrated deep learning framework for solar irradiance prediction, which integrates modern machine learning techniques, including complete ensemble empirical mode decomposition with adaptive noise (CEEMDAN), a Wasserstein GAN (WGAN), and a long short-term memory (LSTM) network. First, the CEEMDAN is leveraged to decompose the original data into several inherent mode functions (IMF) and residual components, dividing the obtained IMFs into high- and low-frequency subsequences for subsequent model prediction. Second, unlike previous similar studies that typically use a single predictor for all IMFs, we consider the spectrum characteristics of the decomposed subsequences in selecting the prediction models. We use a dual channel network structure, the stacked LSTM architecture to predict low-frequency IMFs, and a new stacked WGAN, using a stacked GRU as its generator and an MIP as a discriminator to predict high-frequency IMFs, thus obtaining better modeling. Finally, we combine all the individual results to obtain the final prediction of solar irradiance in the next time step. To sum up, the contributions of this paper are as follows:We propose a new decomposition-integrated deep learning framework for solar irradiance forecasting by integrating modern machine learning methods, including CEEMDAN, WGAN, and LSTM.The irradiance data were decomposed into high- and low-frequency subsequences using CEEMDAN. We predict high-frequency IMFs using the newly customized WGAN and low-frequency IMFs using the stacked LSTM.The proposed model was compared with multiple decomposition integration model and single model structures and the experimental results show that our model has better prediction performance.

## 2. Related Works

Accurate and reliable solar irradiance prediction can significantly benefit the management of power generation and distribution of modern intelligent grids. However, instability, intermittency, and randomness make it challenging to accurately predict solar irradiance. The existing solar correlation prediction methods can be roughly divided into physical-driven and data-driven models based on the mathematical principles adopted [23]. The physical-driven models can use remote sensing and meteorological information to predict future solar power generation. However, when the execution time and computational burden of completing the simulation are unsatisfactory, obtaining the necessary input of physical-driven models on time may be challenging [24]. Data-driven models include traditional statistical, emerging machine learning, and deep learning models. They can improve prediction accuracy by finding the internal relationships between data and are the most commonly used technologies in photovoltaic power prediction research. Currently, the most popular single model structures include convolutional neural networks (CNN), LSTM, GAN, GRU, etc. [25].

The currently popular GAN can effectively overcome the problem of error accumulation in the prediction field, thereby improving prediction accuracy. A GAN has essential applications in capturing the implicit relationships among complex nonlinear sequence data. In [26], a new GAN model for wind power generation forecasting was proposed and the accuracy of the WGAN-GP model for forecasting was verified. Wang et al. [27] used a GAN to enhance the training datasets of each weather type. Finally, more accurate results were obtained by training the CNN on the enhanced dataset. However, training the GAN model to achieve Nash equilibrium is challenging. It has problems such as slow convergence, mode collapse, low sample quality, and an unstable training process [28].

The single model structure also has many shortcomings in processing complex data features, including difficulty in determining super parameters, resource consumption, and slow computing speed [29]. Therefore, scholars focus on hybrid models to improve the accuracy of DL networks [30]. Hybrid models combine multiple models to solve the limitations of single models and use the characteristics of multiple models to improve performance. A mainstream hybrid model is a combination of data decomposition methods and prediction models. Signal decomposition is an essential preprocessing step for building these hybrid models. EMD, VMD, and WTD are typically used for irradiance prediction or radiation prediction. Pi et al. [31] proposed a model based on WTD and a CNN and combined it with the attention LSTM model for the prediction of solar irradiance. Jin et al. [32] proposed a new hybrid artificial intelligence-enhanced prediction model for energy consumption time-series prediction, which combined singular spectrum analysis (SSA) and parallel LSTM neural networks. Zeng et al. [33] proposed a PM2.5 air-quality prediction model that combined an extended stationary wavelet transform (ESWT) and nested LSTM neural networks. Lan et al. [34] used the EEMD algorithm to preprocess the solar irradiance data and self-organizing mapping and back-propagation neural network models to predict the day irradiance. The experimental results showed that this decomposition method can improve prediction performance. Mohammadi et al. [35] proposed combining SVM and WD algorithms to predict solar irradiance. The SVM was used to predict the subsequences of WD decomposition. A hybrid model based on a multi-branch hybrid structure consisting of a 1D convolution and LSTM was used for irradiance prediction. The model used WPD to decompose irradiance and predict subsequences.

In addition to the above-mentioned studies, numerous scholars have devoted their research efforts to studying solar irradiance prediction through the decomposition-integration method. This article follows suit by combining CEEMDAN, WGAN, and LSTM into the decomposition-integration method for further exploration and verification.

## 3. Methodology

The general flow chart of the short-term solar irradiance prediction framework proposed in this paper is shown in Figure 1. CEEMDAN is used to decompose and preprocess the irradiance data, and then the decomposed subsequences are divided into high- and low-frequency subsequences. The high-frequency subsets are input to the WGAN and the low-frequency subsets are input to the LSTM network for prediction. Finally, the prediction results are accumulated. This chapter introduces the principles of the relevant models in detail.

### 3.1. CEEMDAN

Complete ensemble empirical mode decomposition with adaptive noise (CEEMDAN) was developed from EMD, EEMD, and CEEMD. The EMD-based method can adaptively decompose the original data into several IMFs and residuals with different frequencies and scales. They are applied to nonlinear and non-stationary time-series data and are widely used in various fields, including satellite signal analysis, speech recognition, economic data prediction, image processing, etc. EMD has excellent advantages in dealing with non-stationary and nonlinear signals but there is still a problem of “mode mixing.” Mode mixing refers to similar oscillations in different modes or amplitudes in one mode. The EEMD algorithm eliminates the mode mixing in the EMD algorithm by adding white Gaussian noise to the signal [36]. However, the EEMD algorithm cannot eliminate the Gaussian white noise after signal reconstruction, which causes reconstruction errors. In order to overcome the above problems, Flandrin et al. proposed an advanced signal decomposition technology called CEEMDAN [37]. It can eliminate the mode mixing more effectively, produce a reconstruction error of almost zero, and significantly reduce the calculation costs.

Here, we define the *k*-th component obtained from EMD as an operator $E_{k}\left(\right)$ and let $\omega_{i}\left(t\right)$ be the white noise with normal distribution N (0,1). Figure 1a shows the flow chart of the CEEMDAN decomposition and Figure 1b shows the decomposition results of CEEMDAN on the original solar irradiance sequence. The CEEMDAN algorithm steps are as follows:Break down each $X_{i}\left(t\right)=X\left(t\right)+\varepsilon_{0}\omega_{i}\left(t\right)$ using EMD. Then, extract the first IMF, where $\varepsilon_{0}$ is the noise coefficient, i=1,2,…,I, and define the first mode as
(1)$$IMF_{1}\left(t\right)=\frac{1}{I}\sum_{i=1}^{I}IMF_{1}^{i}\left(t\right)$$Calculate the first residual:
(2)$$r_{1}\left(t\right)=X\left(t\right)-IMF_{1}\left(t\right)$$Use the decomposition residuals, and the second mode is
(3)$$IMF_{2}\left(t\right)=\frac{1}{I}\sum_{i=1}^{I}E_{1}\left(r_{1}\left(t\right)+\varepsilon_{1}E_{1}\left(w_{i}\left(t\right)\right)\right)$$Repeat the above steps for each IMF until the residual is obtained. The final residual can be expressed as:
(4)$$R_{k}\left(t\right)=X\left(t\right)-\sum_{k=1}^{K}IMF_{k}\left(t\right)$$
where *K* is the total number of IMFs, and together, the IMFs constitute the characteristics of the original signals on different time scales. The residual clearly shows the trend of the original sequence, which is smoother and the prediction error has been effectively reduced.

### 3.2. LSTM

Sepp Hochreiter and Jürgen Schmidhuber [38] first proposed the long short-term memory network (LSTM), which was carefully designed. Based on an RNN, an input structure is added to each cell state to control the retention of information, thereby addressing the issue of long sequence learning in RNNs. An LSTM network can maintain the long-term memory of neural networks and the model can also be used for solar irradiance prediction. Each cell of an LSTM network has three parts: the forget gate, input gate, and output gate, which, respectively, determine the filtering, preservation, and generation of information. We applied them to predict the low-frequency irradiance-like subsequences following the CEEMDAN decomposition. The complete structure of the model is shown in Figure 2. The LSTM neural network contains multiple gates:

Forget gate. We use the forget gate to determine the information that needed to be filtered out in each IMF component after decomposition. We use the sigmoid function to determine whether to filter out the current input and previous state. The formula is shown in Equation (Equation 5):
(5)$$f_{t}=\sigma \left(W_{f}\cdot \left[h_{t-1},x_{t}\right]+b_{f}\right)$$Input gate. The sigmoid function determines which input information to retain and the tanh updates some parts to become new values. Then, update $C_{t-1}$ to $C_{t}$. The formulas are shown in Equations (6)–(8):
(6)$$i_{t}=\sigma \left(W_{i}\cdot \left[h_{t-1},x_{t}\right]+b_{i}\right)$$
(7)$$\tilde{C_{t}}=tanh\left(W_{c}\cdot \left[h_{t-1},x_{t}+b_{c}\right]\right)$$
(8)$$C_{t}=f_{t}C_{t-1}+i_{t}\tilde{C_{t}}$$Output gate. First, the output part of the unit is determined by the sigmoid function, and then the predicted value points of the model are obtained by multiplying the unit state by the output parts of the tanh and sigmoid gates. The formulas are shown in Equations (9) and (10):
(9)$$o_{t}=\sigma \left(W_{o}\cdot \left[h_{t-1},x_{t}\right]+b_{o}\right)$$
(10)$$h_{t}=o_{t}*\left[tanh\left(c_{t}\right)\right]$$

### 3.3. WGAN

In 2014, Goodfellow et al. [39] proposed the generative adversarial network (GAN) as a breakthrough and a superb idea for machine learning. The GAN is a generative model that is different from a traditional generative model. It avoids learning based on Markov chains and can be trained based on implicit density. The GAN is composed of a generator and a discriminator. The generator generates sample data that conforms to the real data distribution. The discriminator accurately judges and classifies the input information. If the input data are real, the output is 1; If the input data are false, the output is 0. The training of the GAN is divided into two stages: training the discriminator and then training the generator. In training, the two models will constantly update their parameters to minimize their loss functions and output errors.

The traditional GAN model optimizes the training parameters using Jensen Shannon (JS) divergence. The input data of its generator are Gaussian noise and the loss function can be expressed as:(11)$$L_{G}=-E_{z∼P_{g}\left(z\right)}\left[D\left(G\left(z\right)\right)\right]$$
where $L_{G}$ is the loss function of the generator; E[·] is the expected function; G[·] is the generator function; D[·] is the discriminator function; $P_{g}\left[\cdot \right]$ is the noise data distribution; and *Z* is the input noise data vector.
(12)$$L_{D}=-\left(E_{x∼P_{r}\left(x\right)}\left[lgD\left(x\right)\right]+E_{z∼P_{g}\left(z\right)}\left[lg\left(1-D\left(G\left(z\right)\right)\right)\right]\right)$$
where $L_{D}$ is the loss function of the discriminator and $P_{r}\left[\cdot \right]$ is the distribution of the target data *x*. From the above two loss functions, the objective function of the generator–discriminator game can be established, as shown in Equation (Equation 3).
(13)$$L\left(G,D\right)=E_{x∼P_{r}\left(x\right)}\left[lgD\left(x\right)\right]+E_{z∼P_{g}\left(z\right)}[lg\left(1-D\left(G\left(z\right)\right)\right]$$

However, the traditional GAN model uses JS divergence as the model’s loss function, which leads to gradient disappearance and pattern collapse, resulting in unsatisfactory data generated by the generator [40]. Therefore, Arjovsky et al. [40] proposed the Wasserstein generative adversarial network (WGAN) to overcome the above problems. The WGAN uses Wasserstein distance to distinguish the gap between two distributions. When the gap between the two distributions is large, the generator can still be updated to overcome the training problem in the GAN, thereby improving the quality of data generated by the generator and the prediction accuracy. Compared to traditional generation countermeasure network models, WGANs mainly improve the following aspects:The sigmoid activation function of the discriminator output layer is no longer used.The loss function is no longer logarithmic.The value of the discriminator gradient update is controlled between [−c, *c*].Non-momentum-based algorithms such as random gradient descent are used.

Under ideal conditions, Wasserstein distance $W(P_{r},P_{g})$ is continuously differentiable, as shown in Equation (Equation 14).
(14)$$W\left(P_{r},P_{g}\right)=\frac{1}{K}sup_{∥f∥\leq K}\left(E_{x∼P_{r}\left(x\right)}\left[f\left(x\right)\right]-E_{z∼P_{r}\left(z\right)}\left[f\left(z\right)\right]\right)$$
where sup[·] represents the supremum of the function value; *K* is the Lipschitz constant; X=G(z) is the generated data; and ∥f∥≤K means that function f satisfies the K-Lipschitz continuity and function f can be fitted by the neural network. Therefore, this paper uses the WGAN model to train the high-frequency irradiance subsequences after CEEMDAN decomposition, and its model structure is shown in Figure 3. Our WGAN is newly customized. The generator and discriminator are composed of stacked GRUs and MLPs, which are prevalent network structures in DL. They can effectively overcome the gradient problem in neural networks, help stabilize the WGAN model, and improve the model’s prediction performance.

### 3.4. CEEMDAN-WGAN-LSTM

Based on the above theoretical basis, this study constructs a deep hybrid prediction model, namely CEE-W-L, which combines CEEMDAN, WGAN, and LSTM. The overall flow chart of the hybrid CEE-W-L prediction model proposed in this paper is shown in Figure 4. In this section, we introduce the specific process of data training in the model:(1)The single variable solar irradiance data (GHI) are used as input;(2)The maximum and minimum methods are used to normalize the data;(3)CEEMDAN is used to decompose the data sequence into several subsequence IMFs. We divide the first half of the IMFs into high-frequency subsequences and the second half into low-frequency subsequences;(4)The data of each subsequence are divided into a training set and a test set. The training set is used for model training and the test set is used to generate the model prediction results after training;(5)The divided data are input to the built prediction model for training. The high-frequency subsequences are input to the WGAN for prediction and the low-frequency subsequences are input to the LSTM;(6)The final prediction results are obtained by combining all of the prediction results obtained from the high- and low-frequency series and using inverse normalization.(7)Three evaluation indicators (MAE, MAPE, RMSE) are used to evaluate the prediction results.

## 4. Experimental

### 4.1. Data

The dataset used in this study was measured by the National Solar Radiation Data Base (NSRDB) [41], and the data can be accessed at the following link: https://nsrdb.nrel.gov/data-viewer (accessed on 1 November 2022). We collected the irradiance data of Texas for one year from this website. The measuring station is located at 118.31 degrees west longitude and 33.98 degrees north latitude. The climate in this area has four distinct seasons and the daily temperature difference is relatively significant. Therefore, the collected data also have prominent climatic characteristics. The irradiance in summer is relatively stable and the irradiance in winter changes violently. This dataset contains the local solar irradiance measurement data from 1 January to 31 December 2018. The time interval is 5 min, the collection length is 365 days, and there are 105,120 pieces of data. The original dataset was preprocessed and normalized before the experiment. Considering that there was no light at night, we carried out zero clearing. The intensity of solar radiation is different throughout the year, so it was necessary to divide the radiation data into four seasons for independent prediction to make more accurate predictions. We divided the data of each season (90 days, including about 12,960 data samples) into training sets (54 days, including about 7776 data samples), verification sets (18 days, including about 2592 data samples), and test sets (18 days, including about 2592 data samples) in a 3:1:1 ratio.

### 4.2. Experimental Setup

#### 4.2.1. Standardization

In order to eliminate the dimensional impact of solar irradiance data dimensions and improve the prediction model’s calculation speed and prediction accuracy, we normalized the data using the minimum and maximum methods. This article uses the MinMaxScaler function in the Scikit learn 0.24.1 module with Formula (15) to normalize the data, where $X^{\prime }$ represents the normalized data; *X* represents the sample data; and $X_{max}$ and $X_{min}$ are the maximum and minimum values of the dataset.
(15)$$X^{\prime }=\frac{X-X_{min}}{X_{max}-X_{min}}$$

#### 4.2.2. Evaluation Indicators

This experiment selected four evaluation indicators, namely the Mean Absolute Error (MAE), Mean Absolute Percentage Error (MAPE), Root Mean Square Error (RMSE), and R-Squared (R2), to evaluate the performance of the prediction model. These four error indicators were selected not only because they are commonly used indicators in time-series prediction research but also because the selection of these evaluation indicators is more symbolic and persuasive, making the results of this article more acceptable in our view. The calculation formulas of the four evaluation criteria are as follows:(16)$$MAE\left(y,\hat{y}\right)=\frac{1}{n}\sum_{i=1}^{n}|y_{i}-\hat{y_{i}}|$$
(17)$$MAPE\left(y,\hat{y}\right)=\frac{100\%}{n}\sum_{i=1}^{n}|\frac{y_{i}-\hat{y_{i}}}{y_{i}}|$$
(18)$$RMSE\left(y,\hat{y}\right)=\sqrt{\frac{1}{n}\sum_{i=1}^{n}{\left(y_{i}-\hat{y_{i}}\right)}^{2}}$$
(19)$$R2\left(y,\hat{y}\right)=1-\frac{\sum_{i=1}^{n}{\left(y_{i}-\hat{y_{i}}\right)}^{2}}{\sum_{i=1}^{n}{\left(\bar{y_{i}}-\hat{y_{i}}\right)}^{2}}$$
where $y_{i}$ is the result predicted by the model; $\hat{y_{i}}$ is the actual test sample value; and *n* is the total number of test samples. These evaluation indicators are encapsulated in the sk-learn package. They can be called directly, thereby saving time and ensuring that the experiment is carried out more quickly and efficiently.

#### 4.2.3. Experimental Device

The hardware configuration of the experimental server was as follows: a 12th Gen Intel (R) Core (TM) i7-12700H 2.70 GHz processor, an NVIDIA GeForce RTX3050 Laptop GPU, and a win10 operating system. The relevant version numbers required by the program are Python 3.8, Tensorflow 2.3.0, Torch 1.12.1, and Torch Vision 0.13.0. The parameters we set for all the models are listed in Table 1, Table 2, Table 3 and Table 4.

### 4.3. Experimental Analysis

In order to verify the prediction ability of the mixed prediction model proposed in this paper on the assessment of seasonal irradiance, we forecast the four seasonal irradiances in the same region on the same experimental platform using the CEE-W-L model and eight other prediction models. The models included a single RNN model, a single GRU model, a single Transformer model, a single LSTM model, a single WGAN model, a mixed CEEMDAN-LSTM model, a mixed CEEMDAN-WGAN model, and a mixed CEEMDAN-LSTM-WGAN model. The CEEMDAN-LSTM-WGAN model refers to the proposed model, where the subsequence is decomposed by CEEMDAN, the high-frequency sequence is predicted by LSTM, and the low-frequency sequence is predicted by WGAN, herein referred to as CEE-L-W)

Figure 5 shows the solar irradiance data prediction curve of a randomly selected day from the test dataset (subgraphs represent the prediction results of spring, summer, autumn, and winter, respectively). The red line in the figure represents the final prediction results of the model proposed in this paper, the black line represents the real data curve, and the other colored lines represent the prediction results of each comparison model. It can be seen from the trend of the curve that solar irradiance changes rapidly, and the data contain too much high- and low-frequency noise information. Under constantly changing irradiance, it is difficult for a single model to capture the changing trend, and the curve deviates from the true value curve to a large extent. The mixed model based on CEEMDAN decomposition can achieve relatively good prediction results in this case. It can be seen in the prediction fitting diagram that the proposed CEE-W-L model has a better learning ability for the various fluctuations of the GHI data in the four seasons. It is consistent with the actual data on the overall trend and has an accurate prediction effect under the peak, dramatic changes, and fluctuations of the data.

In addition, Table 5 shows a quantitative comparison of the four evaluation indicators for the proposed CEE-W-L model and the eight models mentioned above. From the data shown in the table, it is evident that the overall performance of the four decomposition-integration models (CEE-WGAN, CEE-LSTM, CEE-L-W, CEE-L-W) was better than that of the five single models (RNN, GRU, LSTM, Transformer, WGAN), which shows that the data decomposed by CEEMDAN were more conducive to function fitting and subsequent model convergence. Furthermore, to more intuitively verify the prediction performance of our proposed model, we converted the quantitative evaluation results (MAE, MAPE, RMSE) of the four decomposition-integration models with better performance into a bar chart. As shown in Figure 6, the overall errors of the CEE-W-L and CEE-L-W models were lower than those of the CEE-LSTM and CEE-WGAN models, which further validates the division of the decomposed subsequence into high and low frequencies for dual-channel prediction. In order to verify the above conclusions, we used a Diebold–Mariano (DM) test, and the results are shown in Table 6. In the case of a significant difference, if the DM value is negative, it means that the performance of model 1 was better than that of model 2, and vice versa if the DM value is positive. According to the results in Table 6, the performance of CEE-L-W was significantly superior to that of CEE-WGAN and CEE-LSTM. CEE-W-L performed significantly better than CEE-L-W. Therefore, passing high-frequency signals through the WGAN and low-frequency signals through the LSTM for the four seasons achieved better performance than passing high-frequency signals through the LSTM and low-frequency signals through the WGAN. Compared to the suboptimal model, our proposed model’s MAE, MAPE, and RMSE values decreased by 3.51%, 6.11%, and 2.25%, respectively, for the four seasons. These results sufficiently reflect the advantages of our model.

To sum up, the CEE-W-L method proposed in this paper can achieve the most accurate prediction based on the experimental results shown in the graphs and tables. In the four different statistical verification indicators, the CEE-W-L model outperformed the other benchmark models. Compared with the single LSTM, RNN, GRU, Transformer, WGAN, mixed CEE-LSTM, CEE-WGAN, and CEE-L-W models, we can see that the CEEMDAN-decomposed data were more conducive to function fitting and subsequent model convergence. Compared with the CEEM-LSTM, CEEM-WGAN, and CEE-L-W models, especially after focusing on the CEE-L-W model, we can further conclude that the CEE-W-L model can better preserve and retain the spatiotemporal characteristics of images.

## 5. Conclusions and Discussion

This paper leverages historical data to forecast future patterns of solar irradiance data. The NSRDB public solar irradiance data are used. Based on the study of the data properties and DL models, this paper proposes a combined forecasting model for time-series prediction using data analysis methods and neural networks, which combines the CEEMDAN decomposition model with the WGAN and LSTM prediction models. CEEMDAN is used to decompose the original single variable solar irradiance dataset. Single-column GHI data are converted into multiple subsequence signals and residual signals. Next, the obtained subsequences are divided into high- and low-frequency subsequences, and each subsequence is divided into a training set and a test set for use in the subsequent prediction model. We pass the high-frequency class through the WGAN and the low-frequency class through the LSTM. Then, the prediction results of each subsequence are accumulated to produce the final prediction results. Finally, the prediction efficiency of the CEE-W-L model is evaluated using a fitting curve, error index, and DM test.

According to the experimental results, the proposed method can accurately predict the changes in the real irradiance data, reduce the fitting error to a lower value, and is significantly superior to the other models. In addition, the model in this paper has better and more robust performance than other prediction models and shows superior performance in the four seasons. The prediction methods in this paper are summarized and the following conclusions are drawn. First, for complex, non-stationary data, the waveform decomposition strategy effectively reduces data complexity. Second, the decomposed data can be combined with the updated prediction model. The WGAN has excellent feature extraction ability and can still achieve sound feature extraction efficiency in predicting time-series data with high volatility. Finally, considering the spectral characteristics of the decomposed subsequences, the WGAN has a greater ability to predict high-frequency signals than the LSTM, and the LSTM has higher accuracy and faster efficiency in predicting data with low complexity. We divide their work and use them in cooperation to obtain better modeling. The new proposed prediction framework can effectively support the deployment of photovoltaic power generation systems, which is crucial to developing intelligent grid systems.

The model in this paper uses multiple attempts to select the model parameters, which has limitations in parameter optimization. Future works will include optimizing the model parameters to improve the applicability of the proposed model. 

## Figures and Tables

**Figure 1 sensors-23-02799-f001:**
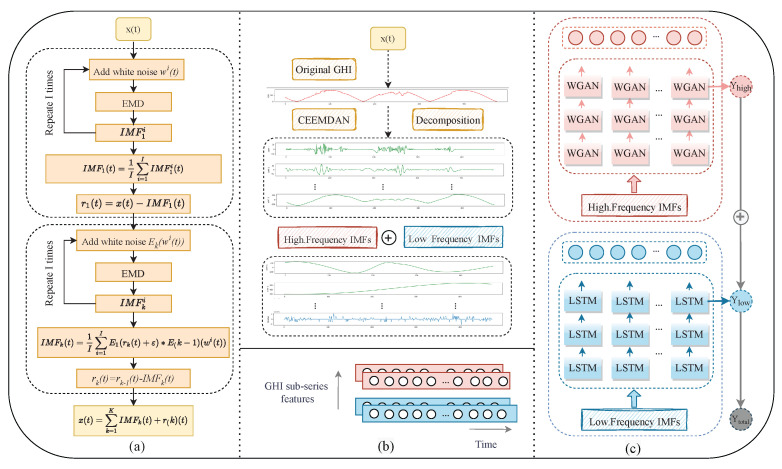
The proposed CEE-W-L overall framework flow chart. (**a**) Flow chart of the signal decomposition module. (**b**) High- and low-frequency characteristic arrangement of GHI data after decomposition. (**c**) Detailed diagram of two-channel neural network prediction.

**Figure 2 sensors-23-02799-f002:**
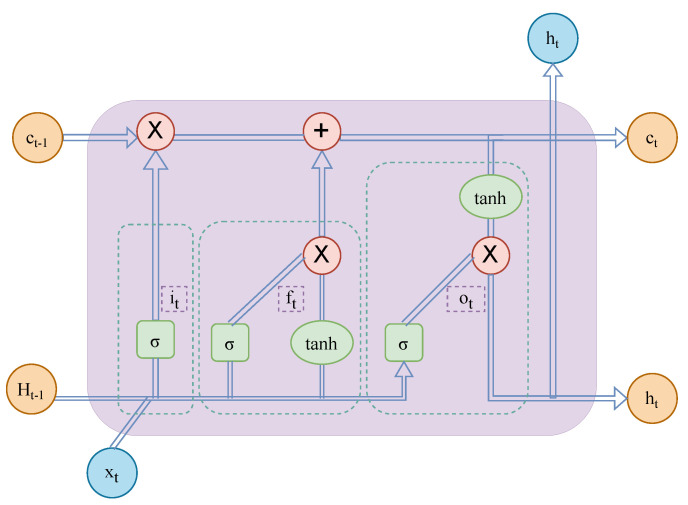
The schematic diagram of the working process of the LSTM neural network.

**Figure 3 sensors-23-02799-f003:**
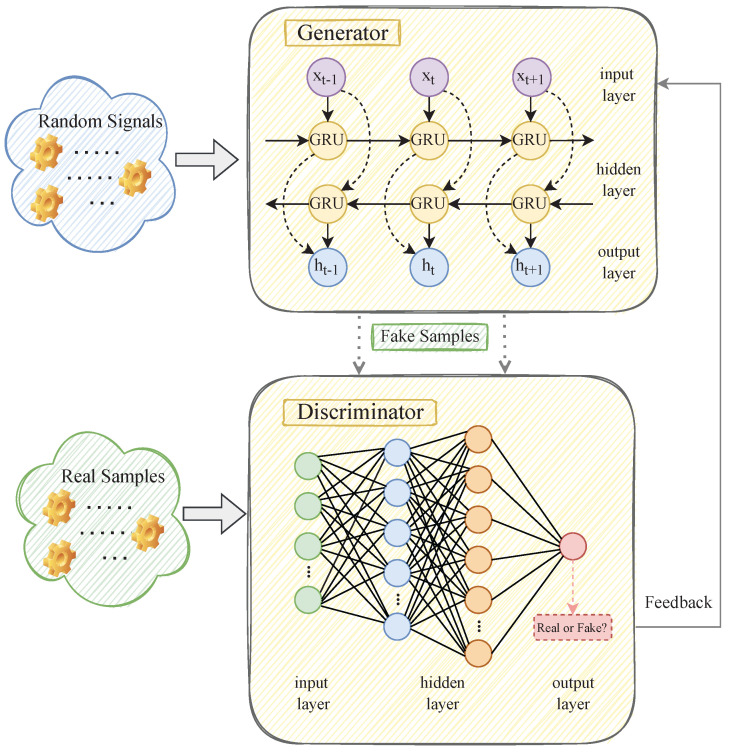
The schematic diagram of the working process of the WGAN neural network.

**Figure 4 sensors-23-02799-f004:**
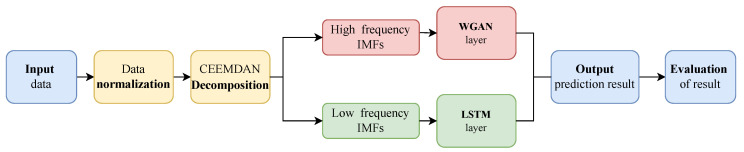
The overall flow chart of all proposed GHI prediction frameworks.

**Figure 5 sensors-23-02799-f005:**
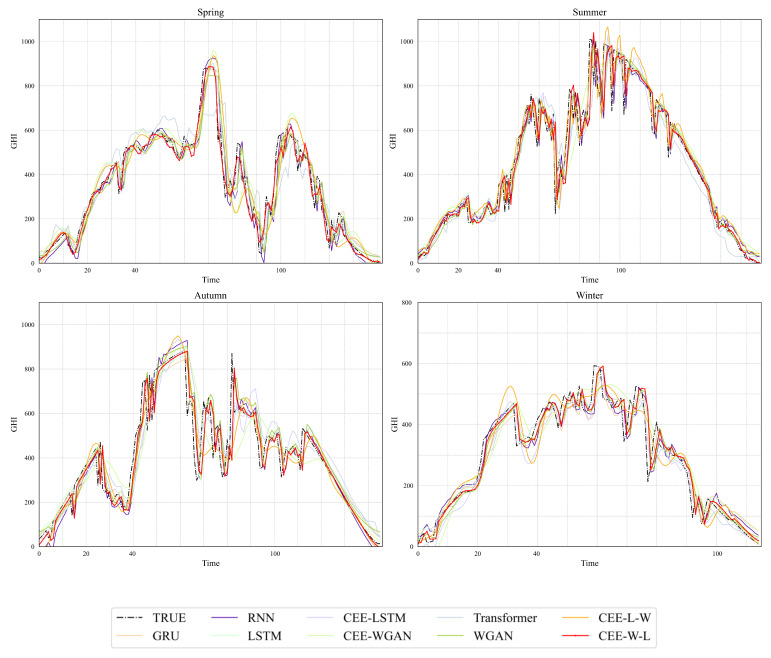
The comparison chart of the data of the four seasons using the different methods. The four graphs in the figure represent spring, summer, autumn, and winter, respectively.

**Figure 6 sensors-23-02799-f006:**
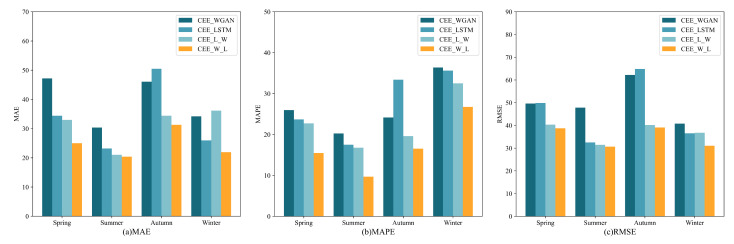
(**a**) Comparison of the MAE of the proposed model and that of the other decomposition-integration models. (**b**) Comparison of the MAPE of the proposed model and that of the other decomposition-integration models. (**c**) Comparison of the RMSE of the proposed model and that of the other decomposition-integration models.

**Table 1 sensors-23-02799-t001:** Required parameters in RNN and GRU.

Parameters	Descriptions	Values
Ahead-num	How long are the historical data used to predict the future	10
Num-layers	Number of stack layers	3
Optimizer	Minimizes the loss function by training and optimizing the parameters	RMSProp
Epoch	Indicates how many forward calculations and back propagations have been completed	100

**Table 2 sensors-23-02799-t002:** Required parameters in Transformer.

Parameters	Descriptions	Values
Num-layers	Number of stack layers	3
Learning-rate	Controls the speed at which we adjust the weights of the neural network based on the loss gradient	0.001
Batch size	Number of data passed to the program for training at a time	32
Optimizer	Minimizes the loss function by training and optimizing the parameters	RMSProp
Drop-out	In the process of training batches, the phenomenon of overfitting can be significantly reduced by ignoring the general feature detector	0.1

**Table 3 sensors-23-02799-t003:** Required parameters in LSTM.

Parameters	Descriptions	Values
Num-layers	Number of stack layers	2
Hidden-size	Number of features in the hidden state	4
Batch size	Number of data passed to the program for training at a time	64
Optimizer	Minimizes the loss function by training and optimizing the parameters	Adam
Epoch	Indicates how many forward calculations and back propagations have been completed	100
Learning-rate	Controls the speed at which we adjust the weights of the neural network based on the loss gradient	0.005

**Table 4 sensors-23-02799-t004:** Required parameters in WGAN.

Parameters	Descriptions	Values
Time steps	How long are the historical data used to predict the future	10
Forecast	How long into the future to predict	1
Hidden-size	The number of features in the hidden size	128
Learning-rate	Controls the speed at which we adjust the weights of the neural network based on the loss gradient	$1\times 10^{-4}$
Batch size	Number of data passed to the program for training at a time	64
Optimizer	Minimizes the loss function by training and optimizing the parameters	RMSProp
Epoch	Indicates how many forward calculations and back propagations have been completed	300
Clipping-value	After each update of the discriminator parameters, it truncates their absolute values to no more than a fixed constant	0.001
N-critic	Number of iterations of the critic per generator iteration	5

**Table 5 sensors-23-02799-t005:** Quantitative comparison of the four evaluation indicators of the proposed model and eight comparison models for four seasons.

		RNN	GRU	LSTM	WGAN	Transformer	CEE-LSTM	CEE-WGAN	CEE-L-W	CEE-W-L
Spring	MAE	38.89	38.34	40.10	40.70	40.85	34.43	47.20	32.97	**24.98**
	MAPE	31.56	34.26	32.31	32.63	26.35	23.67	25.96	22.71	**15.45**
	RMSE	61.43	58.70	51.01	59.57	51.00	49.85	49.60	40.33	**38.75**
	R2	0.956	0.959	0.957	0.942	0.956	0.97	0.949	0.958	**0.982**
Summer	MAE	28.93	26.09	28.91	37.11	28.16	23.20	30.38	22.01	**21.39**
	MAPE	20.79	19.49	23.21	23.89	18.36	17.51	20.23	16.77	**9.67**
	RMSE	47.25	44.32	46.48	54.29	45.70	32.52	47.82	31.44	**30.60**
	R2	0.979	0.981	0.983	0.975	0.982	0.99	0.977	0.986	**0.995**
Autumn	MAE	50.45	40.54	39.00	51.80	49.17	50.48	46.04	34.40	**31.27**
	MAPE	30.37	28.68	29.64	36.71	27.49	33.38	24.16	19.93	**16.53**
	RMSE	56.13	66.89	67.17	66.82	64.09	64.86	62.25	40.15	**39.10**
	R2	0.923	0.94	0.941	0.93	0.928	0.926	0.932	0.935	**0.957**
Winter	MAE	34.32	34.61	34.41	31.16	24.20	25.96	34.17	29.65	**21.92**
	MAPE	50.08	41.97	45.04	42.08	35.25	35.62	36.38	32.51	**26.74**
	RMSE	40.77	44.52	40.77	43.94	38.40	36.52	40.79	36.73	**31.01**
	R2	0.953	0.95	0.955	0.952	0.96	0.964	0.943	0.946	**0.976**

**Table 6 sensors-23-02799-t006:** Results of DM test.

Model 1	Model 2	DM Value	*p*-Value
CEE-L-W	CEE-WGAN	−9.85	0.0000
	CEE-LSTM	−3.192	0.0001
CEE-W-L	CEE-WGAN	−11.67	0.0000
	CEE-LSTM	−5.64	0.0000
	CEE-L-W	−5.37	0.0000

## Data Availability

The source code and required data sets of the experiments can be obtained upon requests.

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
