# Peer review of "A Solar Irradiance Forecasting Framework Based on the CEE-WGAN-LSTM Model"

_sensors, 2023, doi:10.3390/s23052799_

Round 1
Reviewer 1 Report
Comments:
This manuscript seems very interesting despite the topic has already been dealt with in the literature.
I have some suggestions to improve the quality of the manuscript:
1) The abstract should be restructured, i.e. the last sentence should appear in the second sentence and the second sentence may be divided into two sentences afterward.
2) Numerical results may be presented in the abstract.
3) Improved version of CEEMDAN is available; why not authors considered improved CEEMDAN. Moreover, how authors dealt with drawbacks of CEEMDAN.
4) English language may be improved
5) The whole article may be critically reviewed by the authors for format issues/spacing/sentence case and common mistakes like
A large number of experiments show that (lines 12-13)
May be re-written as
Experiments show that
6) The authors only employed error metrics (RMSE, MAE and MAPE). At least two efficiency metrics are also necessary such as R2, Willmott's index of agreement etc.
7) Parameters of WGAN have been presented in Table form. It is better to include parameters and hyperparameters for all models including used for comparison in Table form
8) In addition to line plots, plots such as scatter may also be included.
9) To validate the comparisons further, it is suggested to the authors incorporating additional information such as a statistic analysis. The use of non-parametric tests to assess the conclusions drawn in an experimental study is becoming important in the soft computing community. The authors could consider the use of the Wilcoxon test and the Diebold-Mariano test.
10) Limitations of this study may also be provided
Author Response
Reviewer 1:
Comments:
This manuscript seems very interesting despite the topic has already been dealt with in the literature.
I have some suggestions to improve the quality of the manuscript:
- The abstract should be restructured, i.e. the last sentence should appear in the second sentence and the second sentence may be divided into two sentences afterward.
Reply: Thanks for the reviewer’s comments. We have rearranged the abstract in the revised version to understand the model's logic better. All modifications of the newer version of the manuscript are marked in red color to the reviewer’s easy reference.
- Numerical results may be presented in the abstract.
Reply: Thanks for the reviewer’s comments. In the revised version, we have added at the end of the abstract the important results we obtained in the experimental stage:
“Compared with the suboptimal model, the MAE, MAPE, and RMSE in the four seasons decreased by 3.51%, 6.11%, and 2.25%, respectively.”
- Improved version of CEEMDAN is available; why not authors considered improved CEEMDAN. Moreover, how authors dealt with drawbacks of CEEMDAN.
Reply: Thanks for the reviewer’s comments. CEEMDAN does have some disadvantages: its mode contains some residual noise, especially high-frequency sequences. Therefore, to overcome this shortcoming, after using CEEMDAN to decompose the original signal into IMF components of different frequencies, we continue to carry out wavelet denoising on the high-frequency sequence with residual noise so that the low-frequency information in the high-frequency components can be well-preserved.
- English language may be improved
Reply: Thanks for the reviewer’s comment. We have carefully revised the language errors and asked a couple of native speakers to proofread the paper.
- The whole article may be critically reviewed by the authors for format issues/spacing/sentence case and common mistakes like
Reply: Thanks for the reviewer's comments. We have changed "A large number of experiments show that" to "Experiments show that " Moreover, we have checked and revised the format issues/spacing/sentence case and common mistakes of the whole article.
- The authors only employed error metrics (RMSE, MAE and MAPE). At least two efficiency metrics are also necessary such as R2, Willmott's index of agreement etc.
Reply: Thanks for the reviewer’s comments. In the revised version, we added the efficiency index R2. R2 has been introduced in 4.2.2. Moreover, the R2 experimental results of nine models in four seasons have been added to Table 5.
- Parameters of WGAN have been presented in Table form. It is better to include parameters and hyperparameters for all models including used for comparison in Table form
Reply: Thanks for the reviewer’s suggestions. In the revised version, we have added three-parameter and hyperparameter tables for RNN and GRU, Transformer, and LSTM-related models. They are respectively shown in Table 1, Table 2, and Table 3.
- In addition to line plots, plots such as scatter may also be included.
Reply: Thanks for the reviewer’s suggestions. In order to better verify the prediction performance of the proposed model, we have added a bar chart in the revised version, as shown in Figure 6. Figure 6 shows that we have converted the quantitative evaluation results (MAE, MAPE, RMSE) of four decomposition-integration models (CEE-WGAN, CEE-LSTM, CEE-L-W, CEE-W-L) with better performance into the form of bar chart. It can be seen intuitively from the figure that the error index of our model is the lowest in the four seasons, which further verifies the effectiveness of the proposed model.
- To validate the comparisons further, it is suggested to the au thors incorporating additional information such as a statistic analysis. The use of non-parametric tests to assess the conclusions drawn in an experimental study is becoming important in the soft computing community. The authors could consider the use of the Wilcoxon test and the Diebold-Mariano test.
Reply: Thanks for the reviewer’s suggestions. In the revised version, we added the Diebold Mariano test of four decomposition-integration models with good performance, and the specific results are shown in Table 6. In addition, the conclusions we can get from the table have been discussed in Section 4.3. All modifications made are marked in red.
- Limitations of this study may also be provided
Reply: Thanks for the reviewer’s comments. In the revised version, we added the limitations of this study:
“The model in this paper uses multiple attempts to select model parameters, which has limitations in parameter optimization. Future works include considering to optimize the model parameters to improve the applicability of the proposed model.”

Reviewer 2 Report
The exponential growth of solar energy utilization necessitates an accurate forecasting of solar power generation. To enhance the forecasting accuracy of solar energy generation, the submitted manuscript proposes a prediction method that incorporates Complete Ensemble Empirical Mode Decomposition with Adaptive Noise Analysis (CEEMDAN), Wasserstein Generative Adversarial Network (WGAN), and Long Short-Term Memory Network (LSTM). Initially, CEEMDAN divides the solar output signal into several relatively straightforward sub-sequences with distinct frequency differences. Then the WGAN model predicts the high-frequency sub-sequences, while the LSTM predicts the low-frequency sub-sequences. Finally, the predicted values of each component are combined to obtain the final prediction result.
This work is an original and innovative contribution to the field and will be suitable for publication after the following revisions are carried out:
Major Revisions
1- Abstract
It would be highly advantageous if the abstract is revised to include the significance of the proposed work in terms of its contribution to the relevant field and a statistical description of the essential findings from the study. A clear and concise abstract is crucial for conveying the essence of the research work to the target audience and for drawing their attention to the study.
2- Introduction
The power production from solar panels is deeply affected by environmental conditions, on Page 1 Line 31, 32 (because the solar radiation intensity is generally affected by a variety of complex factors, such as weather conditions….). Please expand the discussion with reference to recent works such as..
1- https://doi.org/10.1177/0958305X221106618
2- https://doi.org/10.3390/su12020608
AI integrated with renewable sources are the future for sustainable sources. There is a lot of work done in the field of renewable energy and AI collectively. On Page 2 Line 56 and 57 (Scholars usually adopt common DL models, such as recurrent neural network (RNN), LSTM, gated ….) should be supported with the recent articles incorporating LSTM and other Deep Learning models.
1. https://doi.org/10.1016/j.renene.2022.07.136
2. https://doi.org/10.1016/j.csite.2021.101671
3- Figure 1
It would be beneficial for the reader if more detail regarding figure 1 is provide in the manuscript. Perhaps by use of sub-captions in the figure and explanation of the sub captions in the text. The purpose of suggesting sub-captions is to provide a deeper understanding and clearer illustration of the information being presented in the figure. This added level of detail will ensure that the information is effectively conveyed and easily understood by the reader.
4- Figure 5
It would be beneficial to improve the quality of Figure 5 as the current images are indistinct and do not effectively showcase the variations in the results. Additionally, the legends are not easily interpretable. It is suggested to present all the seasons in a single graph, given that the data set encompasses a full year of information for the year 2018.
It appears that Figure 5 may not be necessary, as there are only two potential outcomes from the renewable sources, namely electric or thermal energy. To clarify the information, it might be more appropriate to replace the figure with text explaining these two possible outcomes.
5- Evaluation Indicators
Please elaborate the results obtained from evaluation indicators.
6- Data Set
It would be greatly appreciated if the authors could include the citation for the website used to acquire the dataset and why old data set was used in the study. Additionally, it would be helpful to provide reasoning behind why the data was not recorded using any devices to ensure accurate results and validation.
7- Validation
It would have been beneficial to validate the model on multiple days, beyond the selected data set, to get a more complete understanding of its performance and make a detailed comparison. Also, a graph should be presented to demonstrate the forecasted results with the actual measured value of GHI.
8- Training and testing
The authors should clarify the amount of data used for training and testing the model.
9- Comparison
The absence of comparison between the predicted models and actual values raises the question of how the proposed model is determined to be better than other models based on which criteria.
Minor Revisions
1- Acronyms
Acronyms used in the study should be presented in the tabular form before the references.
2- References
Citations should be in standard form and consistent such as on Page 3 Line 116.
3- Page 3 Line 115
The repetition of the citations should be eliminated.
4- Page 5 Line 172
It appears that there may be a typo in Line 172 with Ek(.). Kindly review and make any necessary corrections.
5- Page 6
The numbering of the gates in the LSTM definition should be revised as the current numbering is causing confusion with the equations that have a similar format.
6- Tables
It is recommended to merge the information present in Tables 2 to 5 into a single table with an additional column to indicate the season. This will help to simplify the presentation of the information and eliminate the need for multiple tables. The new table should be organized in such a way that it clearly and concisely presents the relevant information and makes it easier for the reader to understand the results. By incorporating the season information into the table, the reader will be able to quickly identify trends and patterns in the data, making it easier to draw meaningful conclusions from the research.
Author Response
Reviewer 2:
The exponential growth of solar energy utilization necessitates an accurate forecasting of solar power generation. To enhance the forecasting accuracy of solar energy generation, the submitted manuscript proposes a prediction method that incorporates Complete Ensemble Empirical Mode Decomposition with Adaptive Noise Analysis (CEEMDAN), Wasserstein Generative Adversarial Network (WGAN), and Long Short-Term Memory Network (LSTM). Initially, CEEMDAN divides the solar output signal into several relatively straightforward sub-sequences with distinct frequency differences. Then the WGAN model predicts the high-frequency sub-sequences, while the LSTM predicts the low-frequency sub-sequences. Finally, the predicted values of each component are combined to obtain the final prediction result.
This work is an original and innovative contribution to the field and will be suitable for publication after the following revisions are carried out:
Major Revisions
1- Abstract
It would be highly advantageous if the abstract is revised to include the significance of the proposed work in terms of its contribution to the relevant field and a statistical description of the essential findings from the study. A clear and concise abstract is crucial for conveying the essence of the research work to the target audience and for drawing their attention to the study.
Reply: Thanks for the reviewer’s comments. In the revised version, we have rearranged the contents of the abstract and added significant experimental results so that readers can better understand the research focus of this paper.
All modifications made are marked in red.
2- Introduction
The power production from solar panels is deeply affected by environmental conditions, on Page 1 Line 31, 32 (because the solar radiation intensity is generally affected by a variety of complex factors, such as weather conditions….). Please expand the discussion with reference to recent works such as..
1- https://doi.org/10.1177/0958305X221106618
2- https://doi.org/10.3390/su12020608
AI integrated with renewable sources are the future for sustainable sources. There is a lot of work done in the field of renewable energy and AI collectively. On Page 2 Line 56 and 57 (Scholars usually adopt common DL models, such as recurrent neural network (RNN), LSTM, gated ….) should be supported with the recent articles incorporating LSTM and other Deep Learning models.
- https://doi.org/10.1016/j.renene.2022.07.136
- https://doi.org/10.1016/j.csite.2021.101671
Reply: Thanks for the reviewer’s suggestions. We have downloaded and read several excellent documents recommended by the reviewer and added a few discussion points in the introduction part of the revised version, which is marked in red.
“Iqbal et al. [7] found that the environment's dew point temperature and humidity are inversely proportional to the solar radiation intensity. Mustafa et al. [8] found that many environmental factors (such as dust accumulation, water droplets, bird droppings, and partial shading conditions) have an important impact on the performance of the photovoltaic system. ”
“Scholars usually adopt common DL models, such as artificial neural network (ANN), convolution neural network (CNN), LSTM, gated recursive unit (GRU), and generative adversarial network (GAN) [20,21].”
In addition, we also cited these references in the revised version.
3- Figure 1
It would be beneficial for the reader if more detail regarding figure 1 is provide in the manuscript. Perhaps by use of sub-captions in the figure and explanation of the sub captions in the text. The purpose of suggesting sub-captions is to provide a deeper understanding and clearer illustration of the information being presented in the figure. This added level of detail will ensure that the information is effectively conveyed and easily understood by the reader.
Reply: Thanks for the reviewer’s comments. In the revised version, we have added a detailed explanation of the subheadings in Figure 1 to the title. Figure 1 shows the proposed CEE-W-L overall framework flow chart. Where (a) represents the flowchart of the signal decomposition module; (b) represents the high-low frequency characteristic arrangement of GHI after decomposition; (c) represents the detailed diagram of two-channel neural network prediction.
All modifications made are marked in red.
4- Figure 5
It would be beneficial to improve the quality of Figure 5 as the current images are indistinct and do not effectively showcase the variations in the results. Additionally, the legends are not easily interpretable. It is suggested to present all the seasons in a single graph, given that the data set encompasses a full year of information for the year 2018.
It appears that Figure 5 may not be necessary, as there are only two potential outcomes from the renewable sources, namely electric or thermal energy. To clarify the information, it might be more appropriate to replace the figure with text explaining these two possible outcomes.
Reply: Thanks for the reviewer’s comments.
First of all, we have improved the quality of Figure 5 and re-uploaded it into the revised version to ensure that reviewers and readers can more directly see the changes in the results in the figure. Figure 5 shows the predicted change trend of solar irradiance for one day from the test results of four seasons and put them together. The black line represents the real GHI value, the red curve represents the final prediction result of our proposed CEE-W-L model, and the other color lines represent the prediction results of each comparison model. It can be seen from the curve's trend in the figure that the solar irradiance fluctuates wildly, but the trend of the red line is the closest to that of the black line under the peak, sharp change, and fluctuation of the data. This can prove that our model has better prediction ability to some extent.
Secondly, the reviewer mentioned that presenting all the seasons in a single graph is recommended. Because of our extensive data, achieving the GHI of all seasons in one graph is difficult. Therefore, we choose to put in Figure 5, which represents the changing trend of solar radiation on a specific day in spring, summer, autumn, and winter, which can further prove the robustness of the model.
Finally, to enable the reviewer to judge the prediction performance of our model more intuitively, we have added the bar comparison chart of evaluation indicators of the four seasons' data in Figure 6 in the revised version. Our model has the lowest MAE, MAPE, and RMSE.
5- Evaluation Indicators
Please elaborate the results obtained from evaluation indicators.
Reply: Thanks for the reviewer’s comments.
We rearranged the results obtained from the quantitative evaluation indicators into Table 5. In addition to the original MAE, MAPE, and RMSE, we added the efficiency index - R2 in the revised version. The table compares the prediction performance of data from four seasons in nine different models. The smaller the value of MAE, MAPE, and RMSE, the better the model's performance. On the contrary, the higher the value of R2, the better the model's performance. It can be seen from the table that the value of MAE, MAPE, and RMSE of CEE-W-L is the lowest in four seasons, and the R2 of CEE-W-L is also the highest, with the R2 of summer as high as 0.995. Therefore, our CEE-W-L model has a better prediction effect than the other eight comparative models.
6- Data Set
It would be greatly appreciated if the authors could include the citation for the website used to acquire the dataset and why old data set was used in the study. Additionally, it would be helpful to provide reasoning behind why the data was not recorded using any devices to ensure accurate results and validation.
Reply: Thanks for the reviewer’s comments. We have added the data link used in this study in the revised version:
“The data set used in this study is measured by the National Solar Radiation Data Base (NSRDB) [42], and the data can be accessed through the following link: https://nsrdb.nrel.gov/data-viewer. We collected the irradiance data of Texas for one year from the website.”
Reference:
- Sengupta, Manajit and Xie, Yu and Lopez, Anthony and Habte, Aron and Maclaurin, Galen and Shelby, James. The national solar radiation data base (NSRDB). Renewable and sustainable energy reviews 2018, 89, 51–60.
All modifications made are marked in red.
7- Validation
It would have been beneficial to validate the model on multiple days, beyond the selected data set, to get a more complete understanding of its performance and make a detailed comparison. Also, a graph should be presented to demonstrate the forecasted results with the actual measured value of GHI.
Reply: Thanks for the reviewer's comments. In the actual experiment process, we divided the data set into the training set, verification set, and test set according to the ratio of 3:1:1. Sorry, we did not explain this clearly in the paper. The data of training and verification are collectively referred to as training sets. The data used for training, verification, and testing are entirely separated, and the data used for verification and testing are not included in the training data. In other words, the model does not pre-learn the data of our test part. Therefore, the results obtained from the test data are the same as the effect of selecting data from multiple days beyond the selected data set the reviewer suggested for verification.
In addition, the information shown in Figure 5 in the paper is the fitting diagram between the forecasted results of each model and the actual measured values of GHI. For the reviewer and other readers to see more clearly, we have improved the quality of Figure 5.
8- Training and testing
The authors should clarify the amount of data used for training and testing the model.
Reply: Thanks for the reviewer’s comments. We have added a detailed description of the amount of experimental data in the revised version.
“ Considering that the intensity of solar radiation is different throughout the year, it is necessary to divide the radiation data into four seasons for independent prediction to make more accurate predictions. We divide the data of each season (90 days, including about 12960 data samples) into training sets (54 days, including about 7776 data samples), verification sets (18 days, including about 2592 data samples), and test sets (18 days, including about 2592 data samples) in a 3:1:1 ratio.”
All modifications made are marked in red.
9- Comparison
The absence of comparison between the predicted models and actual values raises the question of how the proposed model is determined to be better than other models based on which criteria.
Reply: Thanks for the reviewer’s comments.
On the one hand, in Section 4.2.2 of the revised version, the experimental evaluation indicators of this study are MAE, MAPE, RMSE, and R2. Table 5 shows the quantitative evaluation values of the nine models. It is not difficult to see that the indicators of the proposed models are the best.
On the other hand, to let reviewers and readers see the prediction performance of each model more intuitively, we have added the bar comparison chart of evaluation indicators of the four seasons data in Figure 6. Comparing our model with several decomposition-integration models with outstanding performance, our MAE, MAPE, and RMSE are the lowest.
In addition, to make this conclusion more convincing, we added Diebold Mariano(DM) test to the four decomposition-integration models, and the results are shown in Table 6. It can be seen from the table that the performance of the proposed model is significantly due to the other three models. (The specific discussion content added has been marked in red in Section 4.3)
Minor Revisions
1- Acronyms
Acronyms used in the study should be presented in the tabular form before the references.
Reply: Thanks for the reviewer’s excellent suggestions. In the revised version, we present the acronyms used in the study in the form of tables before the references.
2- References
Citations should be in standard form and consistent such as on Page 3 Line 116.
Reply: Thanks for the reviewer’s excellent suggestions. We checked all citation formats in the revised version and kept them consistent.
3- Page 3 Line 115
The repetition of the citations should be eliminated.
Reply: Thanks for the reviewer’s excellent suggestions. In the revised version, we have deleted duplicate citations.
4- Page 5 Line 172
It appears that there may be a typo in Line 172 with Ek(.). Kindly review and make any necessary corrections.
Reply: Thanks for the reviewer’s excellent suggestions. In the revised version, We have modified “Ek(.)” to “Ek()”.
- Page 6
The numbering of the gates in the LSTM definition should be revised as the current numbering is causing confusion with the equations that have a similar format.
Reply: Thanks for the reviewer’s excellent suggestions. We modified the numbering form of the gates in the LSTM definition in the revised version.
6- Tables
It is recommended to merge the information present in Tables 2 to 5 into a single table with an additional column to indicate the season. This will help to simplify the presentation of the information and eliminate the need for multiple tables. The new table should be organized in such a way that it clearly and concisely presents the relevant information and makes it easier for the reader to understand the results. By incorporating the season information into the table, the reader will be able to quickly identify trends and patterns in the data, making it easier to draw meaningful conclusions from the research.
Reply: Thanks for the reviewer’s excellent suggestions. In the revised version, we have combined Table 2 to Table 5 into one table, as shown in Table 5.
All modifications made are marked in red.

Round 2
Reviewer 1 Report
The revised manuscript can be published as the authors have incorporated all my comments.
Reviewer 2 Report
The authors have made sufficient changes to the manuscript in response to the concerns of this reviewer. The present version of the manuscript has significant improvements and is therefore recommended for publication.